# Human Adipose Mesenchymal Stromal/Stem Cells Improve Fat Transplantation Performance

**DOI:** 10.3390/cells11182799

**Published:** 2022-09-07

**Authors:** Maria Serena Piccinno, Tiziana Petrachi, Marco Pignatti, Alba Murgia, Giulia Grisendi, Olivia Candini, Elisa Resca, Valentina Bergamini, Francesco Ganzerli, Alberto Portone, Ilenia Mastrolia, Chiara Chiavelli, Ilaria Castelli, Daniela Bernabei, Mara Tagliazucchi, Elisa Bonetti, Francesca Lolli, Giorgio De Santis, Massimo Dominici, Elena Veronesi

**Affiliations:** 1Technopole “Mario Veronesi”, Via 29 Maggio 6, 41037 Mirandola, Italy; 2Plastic Surgery, IRCCS Azienda Ospedaliero-Universitaria Sant’Orsola di Bologna, Via Massarenti 9, 40138 Bologna, Italy; 3Dipartimento di Medicina specialistica (DIMES), University of Bologna, Via Massarenti 9, 40138 Bologna, Italy; 4Department of Medical and Surgical Sciences for Children & Adults, University of Modena and Reggio Emilia, Hospital of Modena, Via del Pozzo, 71, 41125 Modena, Italy; 5Rigenerand Srl, Via Maestri del Lavoro 4, 41036 Medolla, Italy; 6Clinical and Experimental Medicine PhD Program, University of Modena and Reggio Emilia, 41125 Modena, Italy; 7Hematological Diagnostics and Clinical Genomics Unit, Modena University Hospital, Via del Pozzo 71, 41125 Modena, Italy

**Keywords:** adipogenesis, adipose stem cells, mesenchymal stem cells (MSCs), tissue engineering

## Abstract

The resorption rate of autologous fat transfer (AFT) is 40–60% of the implanted tissue, requiring new surgical strategies for tissue reconstruction. We previously demonstrated in a rabbit model that AFT may be empowered by adipose-derived mesenchymal stromal/stem cells (AD-MSCs), which improve graft persistence by exerting proangiogenic/anti-inflammatory effects. However, their fate after implantation requires more investigation. We report a xenograft model of adipose tissue engineering in which NOD/SCID mice underwent AFT with/without human autologous AD-MSCs and were monitored for 180 days (d). The effect of AD-MSCs on AFT grafting was also monitored by evaluating the expression of CD31 and F4/80 markers. Green fluorescent protein-positive AD-MSCs (AD-MSC-GFP) were detected in fibroblastoid cells 7 days after transplantation and in mature adipocytes at 60 days, indicating both persistence and differentiation of the implanted cells. This evidence also correlated with the persistence of a higher graft weight in AFT-AD-MSC compared to AFT alone treated mice. An observation up to 180 d revealed a lower resorption rate and reduced lipidic cyst formation in the AFT-AD-MSC group, suggesting a long-term action of AD-MSCs in support of AFT performance and an anti-inflammatory/proangiogenic activity. Together, these data indicate the protective role of adipose progenitors in autologous AFT tissue resorption.

## 1. Introduction

Autologous fat transfer (AFT) has progressively gained an important role in clinical practice and especially in reconstructive surgery [1,2]. Despite the high prevalence of these treatments, fat grafting persistence is very low with resorption rates between 40% and 60% of implanted tissue in addition to long-term atrophy and fibrosis [3,4]. To initially overcome this low performance, multiple repeated AFT procedures are routinely used [5]. However, aside from the complexity and the clinical impact of the procedure, this may not always be feasible due to the lack of adipose tissue (AT) to be harvested and implanted. To counteract these issues and to provide initial insights into AFT resorption, we initially proposed a strategy in which purified AT progenitors, namely, adipose-derived mesenchymal stromal/stem cells (AD-MSCs) [6,7], act as AFT support in a rabbit subcutaneous fat regeneration approach. Cell-assisted lipotransfer (CAL) represents a strategy of fat grafting enhancement, where ex vivo expanded AD-MSCs or stromal vascular fraction (SVF) are added to autologous fat [8]. Several trials have been performed by enriching AFT with SVF to treat lipodystrophy diseases, such as facial atrophy, craniofacial microsomia, and for breast reconstruction in oncology [8,9]. However, CAL by SVF enrichment has major limitations concerning the dose/volume of transplanted cells, possibly limiting the procedure’s consistency [10]. On the other hand, CAL by AD-MSCs may overcome issues of cell dose volume and route of administration, impacting on resorption and volume loss of the graft [10,11].

In a previous study we demonstrated that autologous AD-MSCs ensured both an early protective effect with reduced necrosis and a long-term impact on AFT performance via an increase in angiogenesis [12].

Based on these data and to further investigate the effect of AD-MSC assistance on engraftment efficiency, we here originally developed a murine xenograft model by implanting human AT in combination with human autologous AD-MSCs. Green fluorescent protein-positive AD-MSCs (AD-MSC-GFP) were modified to track their fate following subcutaneous xenotransplant. The same strategy was also performed with AD-MSCs that were not genetically modified to be more consistent with a clinical approach. The protective effect of AD-MSCs on fat grafting was monitored by evaluating both angiogenic and anti-inflammatory effects at 7, 60, and 180 days after transplant, providing new insights into how AD-MSCs promote a better outcome for AFT with relevant clinical impact on patients.

## 2. Materials and Methods

### 2.1. Isolation and Expansion of Human AD-MSCs

Adipose samples (*n* = 4) were harvested according to the Coleman technique [13] during scheduled abdominal liposuctions in patients undergoing breast reconstruction. The study was approved by the local Ethical Committee on Human Study (PROT.N. 3676/CE), and participants provided written informed consent to take part in the study. An average of 600 mg of adipose samples was washed in phosphate-buffered saline (PBS; PAA Laboratories GmbH, Pasching, Austria), minced, and digested for 60 min in Dulbecco’s modified Eagle’s medium (DMEM; Euroclone Spa, Milan, Italy) containing 1.76 UI/mL collagenase solution (Roche Diagnostics GmbH, Mannheim, Germany) and 1% penicillin–streptomycin (P/S; 104 UI/mL and 10 mg/mL; PAA) at 37 °C with gentle agitation [14]. After enzyme inactivation by the addition of blocking media, the resulting cell suspension was centrifuged, filtered through a 100 μm cell strainer (BD Falcon, Durham, NC, USA), counted using 0.4% trypan blue (Biochrom AG, Berlin, Germany), and seeded in culture flasks (Greiner Bio-One GmbH, Frickenhausen, Germany) at a density of 100,000 cells/cm^2^ in Quantum 333 culture medium (PAA) as reported [11]. Once an 80–90% confluence was reached, cells were detached with 0.05% trypsin/0.02% EDTA (Euroclone), counted, and seeded at a density of 6000 cells/cm^2^.

### 2.2. Fluorescence-Activated Cell Sorting Analyses (FACS)

Immunophenotype analysis was performed on the AD-MSC samples after 4 passages labeled with the following monoclonal antibodies: FITC-conjugated anti-human HLA-DR (BD), PE-conjugated anti-human CD31 (BD), CD105 (ebioscience, Waltham, MA, USA), CD73 (BD), PerCP-conjugated anti-human CD45 (BD), APC-conjugated anti-human CD90 (ebioscience), CD14 (BD), CD146 (BD), and unconjugated anti-human GD2 (BD) with an APC-conjugated secondary antibody (BD), in addition to the appropriate isotype controls (BD Pharmigen and BD). The AD-MSCs were analyzed with a FACSARIA III equipped with BD CellQuest Pro^TM^ software (BD, Franklin Lake, NJ, USA), and 10,000 events were acquired.

### 2.3. Differentiation Assays

Cultured AD-MSCs (P4) were tested for their ability to differentiate into bone, adipose, and chondrogenic lineages as previously described [14,15]. Cultured AT-MSCs were seeded in bone induction medium with DMEM (Euroclone) containing 10% FBS (Hyclone, Logan, UT, USA), 1% P/S, and 2 mM glutamine (Euroclone, Padmington, UK) supplemented with 10 nM dexamethasone, 0.1 mM L-ascorbic acid-2-phosphate, 10 mM beta-glycerol phosphate (all Sigma Aldrich, St Louis, MO, USA), and 100 ng/mL bone morphogenic protein BMP2 (PeproTech, Rocky Hill, NJ, USA). After 2 weeks of induction, differentiated AT-MSCs and controls were stained with 1.5% Alizarin Red S (v/v; Sigma) in ddH_2_O. The AT-MSCs were induced towards the adipogenic lineage using DMEM with 1% P/S, 10% rabbit serum (Euroclone), and 5% horse serum (Hyclone) supplemented with 1 μM dexamethasone, 60 μM indomethacin, 10 μM rh-insulin, and 0.5 mM isobutylmethylxanthine (all from Sigma). The AT-MSCs were maintained in differentiation media for 10 days, and adipocyte differentiation was visualized with 1% Oil Red O (Sigma). After induction, the differentiated cells and controls were microscopically visualized using an Axio Observer A1 with an Axiocam MRC5 color camera and Carl Zeiss^TM^ AxioVision 4.8.2 software (Zeiss, Wetzlar, Germany).

To test for chondrogenic differentiation, AT-MSCs were plated in 15 mL conical tubes (2 × 10^5^ cells/mL) in DMEM 4.5 g/l glucose supplemented with 500 ng/mL bone morphogenic protein-6 (Tebu-Bio, Magenta, Milan, Italy), 10 ng/mL transforming growing factor-β (Tebu-Bio), 50 mg/mL ITS + Premix (containing 6.25 μg/mL insulin, 6.25 μg/mL transferrin, 6.25 ng/mL selenous acid, 1.25 mg/mL BSA, and 5.35 μg/mL linoleic acid; BD Biosciences), dexamethasone (final concentration: 100 nM), 0.2 mM L-ascorbic acid-2-phosphate, 100 μg/mL sodium pyruvate (×100), 40 μg/mL proline, 100 nM glutamine (×100), and 1% P/S (all from Sigma). The cells were centrifuged to the bottom of the 15 mL conical tubes and kept in an incubator with a controlled atmosphere (5% CO_2_, 37 °C); the medium was changed every 2 days, leaving the pellet undisturbed inside the tube. At 21 days of differentiation, the pellets were harvested, the formalin was fixed, and the paraffin embedded. Four-micrometer-thick sections of induced samples and negative controls were then specifically stained with 1% Alcian blue solution (Sigma Aldrich).

### 2.4. Prostaglandin E2 (PGE2) Evaluation in AD-MSCs

AD-MSCs (P4) were seeded at a density of 10,000 cells/cm^2^ for 3 days and then stimulated with 8 µg/mL of lipopolysaccharides (LPS, Sigma Aldrich) for 24 h or not for the negative control [16]. Supernatants were collected and analyzed with the Prostaglandin E_2_ Parameter Assay Kit (R&D system, Minneapolis, MN, USA) to quantify PGE2 secretion according to the manufacturer’s instructions. The optical density at 450 nm was measured using an Enspire multiplate reader (PerkinElmer, Whaltman, MA, USA).

### 2.5. Retroviral Transduction of Human AD-MSCs to Express GFP

A bicistronic murine stem cell virus-derived retroviral vector encoding for GFP was used for retroviral transduction of AD-MSCs. Retrovirus production was performed by the FLYRD18 packaging cell line as published previously [14]. The AD-MSCs were then transduced with virus-containing media from FLYRD18-GFP (∼1 × 10^6^ transducing units/mL), as previously described [10], to obtain AD-MSC-GFP cells.

### 2.6. Microbiology Test

Mycoplasma detection on AD-MSCs, which is mandatory for in-house in vivo studies, was performed using the nested PCR Takara Mycoplasma Detection Kit (Takara Bio Inc., Shiga, Japan) according to the manufacturer’s instructions.

### 2.7. Human AD-MSC-Assisted Autologous Fat Transplantation in a Xenogeneic Model

For the xenotransplant experiments, one out of four biological samples was considered also accounting for the complexity of in vivo combining human AFT with autologous AD-MSCs that were additionally gene modified by GFP. A thirty-year-old female patient was recruited for the study after informed consent. The first liposuction was performed using the Coleman technique [13] during a scheduled abdominal liposuction for breast reconstruction. AT was processed and isolated cells were amplified until P2 and cryopreserved for the in vivo implantation together with the autologous fat obtained from the same patient during the scheduled abdominal liposuction. Three weeks before the planned liposuction, the cells were thawed and expanded in culture to obtain the number of cells necessary for transplantation into mice (7 × 10^6^ cells/each group). Six-week-old female NOD.CB17-PrkDCSCID/SZJ mice (*n* = 49; Charles River, Milan, Italy) were maintained in the animal facility at the University of Modena and Reggio Emilia. The study protocol was approved by the local ethical animal committee and by the Italian Ministry of Health. The animals were kept in standard conditions (i.e., 20–24 °C, 30–70% relative humidity, and a 12 h light/dark cycle) for acclimation prior to the study. Animals were divided into 3 groups (*n* = 7 for each time point): 2 groups (i.e., AFT-AD-MSC and AFT-AD-MSC-GFP) received an autologous fat graft supplemented by AD-MSC naive or AD-MSC-GFP and hyaluronic acid (HA), respectively, while a third group received only an autologous fat graft (AFT) resuspended in an HA carrier solution as a control.

The AFT-AD-MSC and AFT-AD-MSC-GFP transplants were prepared immediately after the fat harvest from the recruited patient by gently mixing decanted AT, AD-MSCs, and HA as previously reported [5,12]. For each animal, 1 × 10^6^ AD-MSCs were suspended in 40 μL of carrier solution composed of 25% human serum AB (hsAB), 25% HA hydrogel (Otihyal 1.6, OTI, Carsoli, Italy), 50% saline solution (Laboratori Diaco Biomedicali S.P.A., Trieste, Italy), and then 200 μL of human autologous lipoaspirate was added. For the AFT-AD-MSC and AFT-AD-MSC-GFP groups, each mouse was anaesthetized with 3.6% isoflurane (Schering-Plough Animal Health, Welwyn Garden City, UK); after the preparatory procedures and standard skin sanitization, a graft was injected into the back of the head of each mouse, as previously described [17], using a 1 mL Luer-Lok syringe (BD) connected to an 18 gauge needle (Chemil S.r.l., Padova, Italy). The incision was closed with 9/0 monofilament nonabsorbable sutures (Johnson & Johnson, Auneau, France), and antibiotic ointment (3% Aureomycine, MEDA PHARMA Spa, Milan, Italy) was applied. In the control group (AFT), only 200 μL of decanted AT was injected into the nape of the mice without AD-MSCs.

Animal observation was performed daily for the first week and then weekly by focusing on clinical symptoms analysis (vital signs, appearance, presence and extent of any abnormal response, etc.). Body weight and food intake were measured immediately after transplantation and every week until the study’s termination.

First, we monitored the early effect of the treatment, sacrificing all animal groups at 7 days. Given the importance of the different mechanism of MSC localization and their impact on the therapeutic effects, we also studied the long-term outcome. In particular, we monitored the cell localization and morphology (adipocyte-like versus a fibroblast shape) of the AD-MSC group within the graft until 60 days post-transplant and the therapeutic/biological effects until 180 days in nonengineered AD-MSCs.

In summary, animals were sacrificed according to the following time points: 7 and 60 days post-transplant for the AFT-AD-MSC-GFP group; 7 and 180 days post-transplant for the AFT-AD-MSC group; 7, 60, and 180 days post-transplant for the AFT control group.

At the end of each time point, animals were sacrificed using an overdose of CO_2_ anesthesia, and subsequent analyses were performed.

### 2.8. Hematology and Biochemistry

Peripheral blood was collected by both retro-orbital sinus bleeding and facial vein sampling under anesthesia. For hematological analysis, 200 μL of blood samples were collected in 1.5 mL tubes precoated with 50 μL of 50 UI/mL sodium heparin (Hospira Italia SrL, Napoli, Italy). The samples were then diluted at 1:1 with a saline solution and analyzed using an XE-2100 hematology analyzer (Sysmex, København S, Denmark) within 12 h of sampling. For biochemical analyses, 500 μL of blood was collected in 1.5 mL tubes and kept for 30 min at room temperature to form clots. Samples were centrifuged at 3200 rpm for 15 min, and the supernatant serum was transferred to new tubes and stored at −20 °C. The thawed sera were analyzed for creatinine dosage and aspartate transaminase (AST) activity (from whole sera and diluted 1:1 sera, respectively) with a spectrophotometer (Cobas C501, Roche Diagnostic). In addition, PGE2 levels were analyzed with a Prostaglandin E_2_ Parameter Assay Kit (R&D system) to quantify their levels in AFT and AFT-AD-MSC sera. The optical density at 450 nm was measured using an Enspire multiplate reader (PerkinElmer).

### 2.9. Tissue Handling and Histology

After sacrifice, nape transplants were harvested and the spleen, liver, kidneys, heart, lungs, ovaries, and brain of each animal were collected and weighed. Histological analyses were performed by hematoxylin and eosin staining (H&E) (Carazzi Hematoxylin and Eosin, Carlo Erba Reagenti SpA, Milan, Italy).

The grafts were observed by an Axiozoom V16 microscope (Zeiss) equipped with ZenPro software (Blue edition, Zeiss, Munich, Germany). Seven representative sections were acquired at low magnification (i.e., 1.5× with a digital magnification of 270×) in order to characterize the adipose tissue through the analysis of the percentage area related to adipocytes as well as the count of the adipose cells. To detect blood vessels, angiogenesis, and inflammation in xenotransplanted tissues, immunohistochemistry using anti-CD31 and anti-F4/80 antibodies was performed as previously described [18,19]. Stained slides were then examined using a Zeiss Axioskop (Zeiss), and photomicrographs were acquired using an Axiocam IcC3 color camera and AxioVision 4.8.2 software visualization (Zeiss).

Scoring was performed by counting CD31-positive cells/100× high-power field (*n* = 10 per stained slide) as well as for F4/80-positive cells. High-power fields were picked in the external area of the graft corresponding to the connective capsula. The percentages of pixel-positive F4/80 staining were measured as a percentage of the positive area/100× high-power field (*n* = 10 per stained slide) using Image J software analysis [12].

### 2.10. Statistical Analysis

Data are expressed as the mean values ± standard error of the mean (SEM). Statistical significance was determined by a two-tailed Student’s *t*-test. A *p*-value of <0.05 was used to define statistical significance.

## 3. Results

### 3.1. AD-MSCs Could Be Isolated and Genetically Modified from Small Amounts of Adipose Tissue

The AD-MSCs were isolated from an average of 599.25 ± 135.03 mg of AT, and the digested stromal vascular fraction (SVF) was seeded in culture flasks; fibroblastoid elements began to adhere 48 h after seeding and reached confluence in 8 days (Figure 1a). These adherent cells were cultured until P4 and showed a cumulative doubling of the population of 4.50 ± 0.16 (Figure 1b). To define the nature of the isolated cells, we phenotypically characterized them using FACS analyses: CD14, CD45, HLA-DR, CD31, CD146, CD90, CD105, and CD73. As seen in Figure 1c, the cells displayed physical parameters typical of a stromal cell population, being negative for CD14, CD45, HLA-DR, CD31, and CD146 and positive for CD90 (98.98 ± 0.57), CD105 (98.99 ± 0.51), and CD73 (98.75 ± 0.53).

Multilineage potentials of isolated AD-MSCs were assessed at P4 by inducing differentiation towards osteogenic, adipogenic, and chondrogenic lineages. After 14 days of osteogenic induction, calcium deposits positive to Alizarin Red were largely found in induced samples but absent in noninduced controls (Figure 1d). Similarly, adipogenic induction was able to generate round lipid droplets inside the cells positive to Oil Red O staining (Figure 1d). Finally, after 21 days of chondrogenic differentiation, pellet culture showed extracellular matrix formation positive for Alcian blue staining (Figure 1d).

Additionally, AD-MSC populations were transduced with a vector encoding for GFP to track human adipose-derived cells and to distinguish them from grafted fat tissue after xenotransplant. As seen in Figure 1e, the majority of the AD-MSC population was GFP positive (>98%).

### 3.2. Xenotransplantation Was Well Tolerated by NOD/SCID Mice

The AD-MSCs were isolated from the AT after liposuction collected according to the Coleman technique [13]. After in vitro expansion, AD-MSCs (GFP modified and control) were embedded in a defined mixture of HA, hsAB, and autologous AT obtained from the same patient during a second liposuction procedure. Grafts (approximately 200 mL each) were then subcutaneously transplanted in NOD/SCID mice. The control group received only AT (AFT) from the same subject (Figure 2).

Xenotransplantation was well tolerated. No infection or serum accumulation was detectable on the graft sites, and no clinical or behavioral abnormalities were observed during the weekly surveillance over a 60-day period. The AFT-AD-MSC-GFP treated mice showed a similar trend in body weight average compared to the AFT controls (Figure 3a), and no differences in food intake were observed between the two groups (Figure 3b).

To further explore the safety of our transplantation approach, serological and hematological assays on blood samples from the mice were performed. As shown in Figure 3b, creatinine levels were within the expected range [20]. Curiously, higher levels of CREJ2 were detected in the AFT group compared to AFT-AD-MSC-GFP, both at 7 and 60 days, while no statistically significant differences were observed in aspartate aminotransferase level (ASTL) values, which again persisted in the ranges [15].

Both at 7 and 60 days post-transplantation, there were no graft-related effects on the principal hematological parameters including white blood cells, red blood cells, hemoglobin, platelets, and hematocrit (Figure 3c).

At the end of the two time points (i.e., 7 and 60 days), the mice were sacrificed, the organs were collected and evaluated by macroscopic observation, and their weight was recorded. No abnormalities were noticed among both the AFT-AD-MSC-GFP or AFT groups, which had average organ weights (i.e., liver, spleen, kidneys, ovaries, lung, heart, and brain) that were comparable between treated and control groups at 7 (Figure 3d) and 60 days (Figure 3d) after transplantation (*p* > 0.05). At 7 days post-transplantation, grafts had an average weight of 124.19 ± 18.90 mg in the AFT group and 135.83 ± 29.91 mg in the AFT-AD-MSC-GFP group, with no significant differences in the groups (*p* > 0.05; Figure 3e). Interestingly, at 60 days from transplant, we were able to observe a statistically significant graft weight difference among the AFT and AFT-AD-MSC-GFP groups, with an average of 101.00 ± 7.07 mg and 174.40 ± 15.49 mg, respectively (*p* < 0.05). This suggests that AD-MSC-GFP may contribute to greater persistence in graft mass.

### 3.3. AD-MSC-GFP Differentiated into Adipocytes within 60 Days after Transplantation

To provide further insights into the AD-MSCs influence on increasing graft weight at 60 days, H&E staining was performed. As shown in Figure 4a and Appendix A, both AFT and AFT-AD-MSC-GFP showed a thin layer of connective tissue in the external area and a central core predominantly composed of adipocyte clusters immersed in an extracellular structure, including stromal elements, lipidic cysts, inflammatory cells, and vessels, at 7 days. Similarly, at 60 days, the grafts were both characterized by AT arranged in lobules and surrounded in a thin layer of connective tissue, and lipidic cysts were detected in smaller areas (Figure 4b and Appendix A).

To further confirm these findings, adipose differentiation in the AFT and AFT-MSC-GFP groups was scored with a quantitative approach by advanced microscopy. The number of adipocytes was scored, resulting in an average value (±SD) of 28.4 (±7.6) and 45.6 (±6.73) in the AFT and AFT-AD-MSC-GFP groups, respectively (Appendix A). The addition of AD-MSC-GFP increased the adipocytes number in a statistically significant manner (*p*-value < 0.01), with a fold increase equal to 1.6. In addition, the area corresponding to adipocytes was automatically quantified by ZenPro Software resulting in a percentage average area (±SD) of 20.95 ± 4.42% and 30.40 ± 3.16% in AFT and AFT-AD-MSC-GFP, respectively (Appendix A). The AFT-AD-MSC-GFP groups had a statistically significant increase in adipose tissue area (*p*-value < 0.01). These results further support the evidence that AD-MSC-treated mice retain a higher differentiation potential in comparison to AFT alone.

At each time point, AFT specimens seemed to be more vascularized and less infiltrated compared to AFT-AD-MSC-GFP. To verify this finding, AD-MSC-GFP were tracked by GFP staining (Figure 4c). At day 7, GFP-positive cells with fibroblastoid morphology were detectable in AFT-AD-MSC-GFP specimens in the thin layer of connective tissue among adipocyte clusters. At day 60, the AFT-AD-MSC-GFP specimens showed a significant fraction of GFP-positive adipocytes derived from AD-MSC differentiation that might have contributed to the observed increase in graft weight (Figure 4d). As expected, GFP-positive cells were not detected in the AFT specimens (Figure 4c,d).

### 3.4. Proangiogenic and Pro-Inflammatory Characterization of AFT-AD-MSC-GFP

Having observed GFP-positive adipocytes, we then focused on inflammation and angiogenesis as key elements in tissue resorption at 60 days. At 7 days post-transplantation, the AFT-AD-MSC-GFP specimens had lower numbers of CD31-positive cells compared to the AFT specimens (Figure 5a). The average number of blood vessels was 23.57 ± 1.81 in AFT-AD-MSC-GFP in contrast to 29.32 ± 6.93 in the AFT specimens (Figure 5c, *p* < 0.05). At 60 days post-transplantation, staining showed a smaller quantity of CD31-positive cells in AFT-AD-MSC-GFP (Figure 5b) compared to the AFT specimens (87.2 ± 9.4 and 112.44 ± 12.13, respectively; *p* < 0.05; Figure 5b,d). These data suggest that human AD-MSC-GFP functions after xenotransplantation have a negligible proangiogenic effect in supporting graft persistence in the long term. F4/80 expression, a known macrophage and monocyte marker [19], was subsequently investigated. As shown in Figure 5e, F4/80-positive cells morphologically resembling macrophages were prevalently observed in the connective capsula and tissues among fat lobules. Lipidic cysts also appeared to be positive for F4/80, as reported previously [19]. Expression of F4/80 staining was then quantified using ImageJ software (National Institute of Health, Bethesda, MD, USA) and signals derived from lipidic cysts were excluded. At day 7 after transplantation, the amount of F4/80 positivity was comparable between the two groups (Figure 5e), with an average of 1.92 ± 0.07% compared to 1.63 ± 0.44% for the AFT-AD-MSC-GFP and AFT groups, respectively (Figure 5g, *p* > 0.05). Conversely, at day 60 after transplantation, we observed increased positive signals in the AFT-AD-MSC-GFP group compared to the AFT group with 2.85 ± 0.65 and 2.23 ± 0.30, respectively (Figure 5h, *p* < 0.05), suggesting that AD-MSC-GFP may support greater inflammatory activity linked with a greater macrophage infiltration.

### 3.5. Wild-Type AD-MSCs Preserved Their Proangiogenic and Anti-Inflammatory Properties

To investigate the increased inflammation observed at 60 days using gene-modified AFT-AD-MSC-GFP and accounting for the fact that GFP expression may generate immunogenicity [21] in some models, AFT were also implanted with and without wild-type AD-MSCs, monitored for 7 and 180 days, and compared with the control group treated only with AFT.

At 7 days post-transplantation, grafts had a similar weight among the groups (*p* > 0.05; data not shown). Both the AFT and AFT-AD-MSC specimens shared similar histological features characterized by AT arranged in small lobules immersed in connective tissue that were composed of stromal elements, lipidic cysts, inflammatory cells, and vessels (Figure 6a). Inside the connective tissue, blood vessels and inflammatory cells were differentially visible between the two groups. CD31 staining detected positive cells in both groups (Figure 6b) with a count of 29.32 ± 6.4 and 26.93 ± 2.34 in AFT and AFT-AD-MSC groups (*p* > 0.05), respectively (Figure 6g). Regarding inflammatory cells, the AFT-AD-MSC specimens had decreased F4/80 staining compared to AFT (Figure 6c). Quantification using ImageJ confirmed this initial observation with average F4/80 expression signals of 1.63 ± 0.36 and 1.27 ± 0.48 in the AFT and AFT-AD-MSC groups (*p* < 0.05), respectively (Figure 6g).

Seven days after transplantation, the AFT-AD-MSC group showed an increased number of blood vessels and improved anti-inflammatory activity with a reduction of F4/80-positive cells in grafts treated with wild-type AD-MSCs compared to the AFT group. These findings suggest an early, equal angiogenesis level in both groups and a significant anti-inflammatory impact associated with wild-type AD-MSC transplant.

To further challenge the model, we then assessed the long-term performance of the grafts by monitoring the animals for 180 days. At that time, xenotransplants were well tolerated with a higher average weight in the AFT-AD-MSC group than in the AFT controls (39.28 ± 9.02 mg and 31.33 ± 8.54, respectively; *p* > 0.05). H&E staining revealed that AFT was characterized by the presence of connective tissues with few cellular elements, several immune cells, and lipidic cysts resulting from necrosis (Figure 6d). In contrast, we observed adipocytes organized in small lobules and embedded in a thin layer of connective tissue in AFT-AD-MSC (Figure 6d), which resembled a preserved architecture of the AT. These observations suggest that AD-MSCs play a pivotal role in the long-term persistence of AT in grafts.

These differences were explored to verify whether a better long-term AFT persistence could be attributed to the proangiogenic and/or anti-inflammatory effects of AD-MSCs. The AFT-AD-MSC specimens contained more CD31-positive blood vessels compared to AFT (Figure 6e). The average number of blood vessels counted was 45.7 ± 5.20 compared to 31.07 ± 4.94 in AFT-AD-MSC and AFT (*p* < 0.05), respectively (Figure 6h). These data suggest that in an autologous xenotransplantation model by nonmodified AFT-AD-MSC, angiogenesis is detectable at a long time point following engraftment. The inflammatory activity at 180 days after engraftment was slightly higher in AFT compared to AFT-AD-MSC (Figure 6f), with 1.83 ± 0.51% and 1.49 ± 0.44% of positive area in AFT and AFT-AD-MSC, respectively, as quantified using ImageJ (Figure 6h).

The anti-inflammatory activity and proangiogenic role of AD-MSCs were further investigated [16]. After 24 h of LPS stimulation, the levels of secreted PGE2 were measured, and stimulated AD-MSCs secreted in vitro higher levels of PGE2 (13.062 ± 4.45 pg/mL) compared with unstimulated cells (8.023 ± 1.18 pg/mL) in a statistically significant manner (*p*-value < 0.01; Appendix A).

To better highlight the potential role of PGE2 in mice, we also quantified their levels in the serum of AFT-AD-MSC mice in comparison with only AFT-treated mice after 6 months of transplantation. The circulating levels of PGE2 in AFT and in AFT-AD-MSC treated mice were, respectively, 27.95 ± 11.35 and 54.78 ± 7.76 pg/mL, suggesting that in vivo AD-MSCs can exert their anti-inflammatory/proangiogenic effect in relationship with PGE2 (*p*-value < 0.001; Appendix A).

## 4. Discussion

Although AFT is largely used in the treatment of congenital and acquired soft tissue damage/disorders, the long-term persistence of AT is difficult to predict, probably due to the poor angiogenesis potential in AFT with a high resorption rate [2]. This problem highlights the urgent need for more efficient clinical procedures. As previously demonstrated, the addition of AD-MSCs can improve fat graft survival [7,12], but little is still known regarding the exact fate of the implanted AD-MSCs and the optimal cell load for use in humans [22]. In this study, we investigated the role of human AD-MSCs to potentially increase human AT graft persistence. We were able to isolate 15 × 10^6^ cells/mg of lipoaspirate in 20 days of ex vivo expansion time. We used less than 1 g of AT, in contrast to other published work that reported isolating a comparable number of AD-MSCs from at least 15 mL of lipoaspirate [6,12,17,18]. After the ex vivo expansion of AD-MSCs in serum-free media, an immunophenotypic profile and multipotency assessment of the cells led to a better characterization and a greater homogeneity of the cell population, providing initial information on the effects generated solely by AD-MSCs in the graft.

Previous studies suggested that the water content of lipoaspirate may interfere with the integration of AD-MSCs in AT and lead to a loss of adipose cellular components from the graft site, ectopic fibrosis, and calcification [8,23,24]. To limit ectopic cellular migration, we combined AD-MSCs and AT with HA as a carrier agent, which provided a more physiological microenvironment and improved cell survival and graft persistence. HA hydrogel has been used for cell delivery in several preclinical models of soft tissue and cartilage regeneration and does not impact cell survival or proliferation rate [12,19,25].

We reported a more standardized approach where CAL was performed by administrating a specific dose of AD-MSCs supplemented to a defined amount of lipoaspirate. In addition, we introduced a carrier solution to embed cells and fat. As we previously demonstrated [12], HA exerts the ability to entrap AD-MSCs in the graft, avoiding escape and improving the protective role of AD-MSCs in the graft’s survival.

After the transplantation procedure, we could not detect in vivo toxicity or significant differences in food intake in the hematological/serological parameters or in the animals’ organs, which corresponded with previous reports [19,25,26]. We did not observe clues of hepatic toxicity, having detected a physiologic increment in body weight in the AFT group compared to the AFT-AD-MSC-GFP group as well as an expected increase in the ASTL in the AFT specimens compared to the AFT-AD-MSC-GFP specimens.

Given the biological impact of AD-MSCs in the survival of the graft, we studied cell localization and morphology within the implant at 7 and 60 days.

We first showed that AD-MSCs were detected as fibroblastoid elements in the graft 7 days after transplantation. Sixty days after engraftment, AD-MSC-GFP differentiated in vivo into adipocytes and contributed to an increase in a graft mass of 30% compared to AFT. Furthermore, we detected some small fibroblastoid GFP-positive cells in the connective lobule among AT, suggesting that some AD-MSCs remain in an undifferentiated status, supporting the idea of the persistence of uncommitted AD-MSC-GFP after implants.

Regarding the role of AD-MSC-GFP in grafts, we did not observe proangiogenic or anti-inflammatory effects at either time point. Indeed, in GFP-modified AD-MSCs, we detected increased inflammation, suggesting an immunological response in the NOD/SCID murine model. Others previously described as GFP may be immunogenic in animal models, also indicating how strains and routes of administration may play roles in the immune response of eGFP-modified cells administered in vivo [19,21,26].

In this context, future studies may be performed using other cell tracking approaches such as iron nanoparticles [27].

The group treated with AD-MSC-GFP was introduced with the main aim of tracking human cells in vivo. While evidence exists that GFP labeling does not modify the MSCs’ biological activities [28,29], to be more clinically compatible and to understand the proangiogenic and anti-inflammatory roles in our humanized NOD/SCID model, unmodified AD-MSCs were introduced and observed from 7 to 180 days. Having accounted for the impact of gene-modified cells in AFT, we focused on investigating the long-term outcome of wild-type AD-MSC transplantation in AFT, avoiding GFP as a possible confounding factor and introducing a strategy to be monitored for 180 days. Surprisingly, a greater proangiogenic activity was not detected in the unmodified AD-MSC-treated model at 7 days; this observation was also reported by Stillaert et al. and may be because new blood vessel formation requires at least 14 days after transplantation in an immunodeficient animal model [30]. On the contrary, the decrease in macrophages in AFT-AD-MSC compared to AFT suggests that AD-MSCs retain an anti-inflammatory potential, as also indicated by Bowles et al. [31]. They demonstrated that treatment with SVF and AD-MSCs in an autoimmune encephalomyelitis mice model reduces cell infiltration. In detail, AD-MSCs or fresh SVF were able to attenuate the activities of TH1 and TH17 cells and their associated proinflammatory cytokines. A marked increase in interleukin-10 (IL-10) was correlative to increased regulatory T cells (Tregs). This immunomodulation promotes the differentiation of alternative activated macrophages M2, which produce reparative tissue effects [31]

The anti-inflammatory response though M2 macrophage differentiation is regulated also by PGE2, which is able to favor the resolution phase, the increase in phagocytic activity, and the decrease in inflammation [32]. In addition, PGE2 is reported to be involved in angiogenesis by targeting endothelial cells on EP1-4 receptors and promoting the assembly of new blood vessels through selective activation of the PKACγ pathway [33]. Interestingly, our in vitro data indicate that AD-MSCs’ activation by LPS is followed by an increased PGE2 secretion. PGE2 release by AD-MSCs has previously been reported by others also implicated in an augmented MSC performance [34]. Moreover, the in vivo findings of increased PGE2 levels in the AD-MSC-treated group open an intriguing scenario that shall have to be further investigated.

At 180 days, no residual AT was detected in the AFT group, which suggests a 100% resorption rate of the graft that was substituted with fibrosis. In contrast, in five to six mice treated with AFT-AD-MSC, we observed AT graft persistence with well-preserved architecture and much better vascularization compared to the AFT sample alone. Previous data described how adipose progenitors were a heterogeneous cell source with a fraction able to differentiate in adipocytes, smooth muscle, and endothelial cells [35]. In our findings, only a fraction of AD-MSCs were able to differentiate into adipocytes, resulting in an increase in graft size compared to AFT alone. The remaining fraction of AD-MSCs preserved a fibroblastoid shape, localizing into connective tissue among the adipose lobular areas presumably playing a role in angiogenesis.

Dong at al. implanted AT-derived cells/fragments from a transgenic mouse into a syngeneic C57BL/6J model, indicating that fragmented AT can exert a proangiogenic effect for up to 8 weeks [36]. Collectively, while we acknowledge the relevance of previously reported data on CAL [10,37], we presume that our findings may further support the implementation of progressively more standardized procedures in adipose tissue regeneration.

## 5. Conclusions

In this study, for the first time to our knowledge, we implemented a xenogeneic mouse model where human AFT was enriched by autologous AD-MSCs previously isolated from the same donor. This approach successfully recapitulated human clinical procedures, where AD-MSCs may be introduced to enrich AT, limiting necrosis and fat resorption [8]. While the management of the animal model was cumbersome with multiple transplantations from the same donor, we were able to point out the importance of AD-MSCs and their role in the initial anti-inflammatory phase followed by a later proangiogenic effect in AFT, resulting in AT preservation with potential implications in regenerative surgery.

## Figures and Tables

**Figure 1 cells-11-02799-f001:**
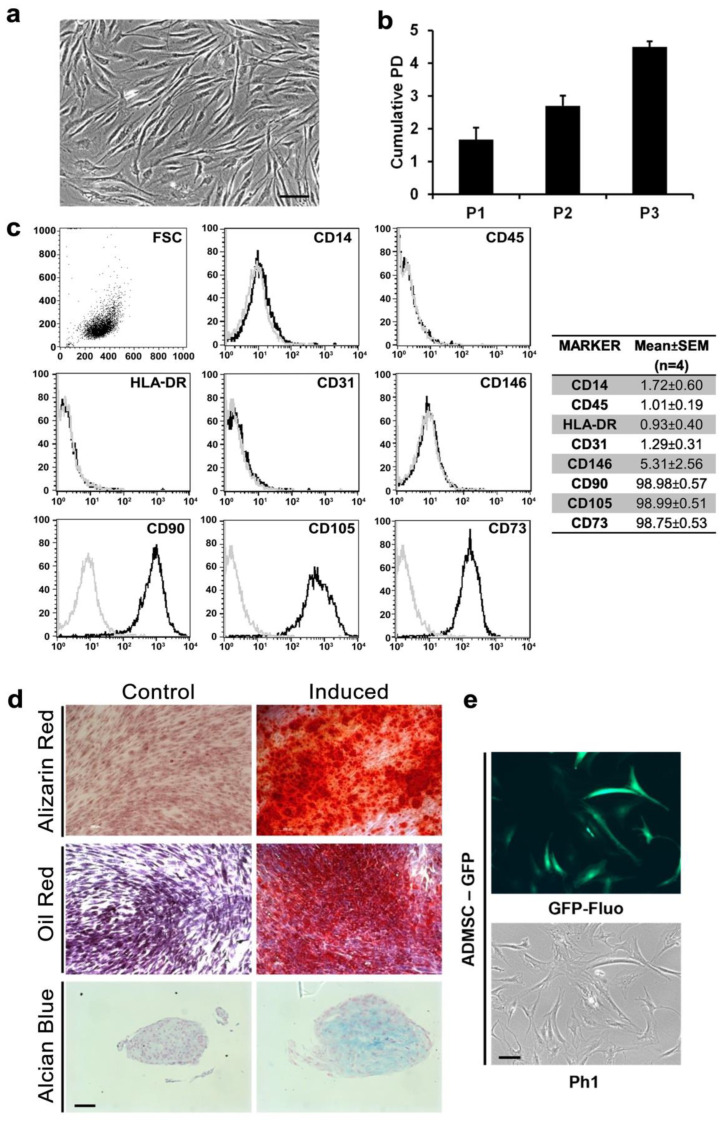
Human lipoaspirate originate performing AD-MSCs in vitro: (**a**) photomicrograph of fibroblastoid elements adhering to plastic after adipose tissue digestion (Ph1, scale bar = 100 μm); (**b**) AD-MSC cumulative population doubling after passage 3 (P3); (**c**) representative immunophenotypical characterizations of AD-MSCs, with cells positive for CD90, CD105, and CD73 and negative for CD14, CD45, HLA-DR, CD31, and CD146 (isotype control in gray; on the right, the table shows the mean values ± SEM of antigen expressed on AD-MSCs by FACS; (**d**) differentiation assays on expanded cells after passage P3; osteogenic differentiation visualized by Alizarin Red staining after 14 days of induction and a corresponding noninduced control (upper panels), adipogenic differentiation after 10 days of induction visualized by Oil Red O staining and a corresponding noninduced control (middle panel), and three-dimensional chondrogenic differentiation after 21 days of induction in pellet culture stained with Alcian blue and a corresponding noninduced control (lower panel), scale bar = 100 μm; (**e**) AD-MSCs expressed GFP after retroviral transduction visualized by both GFP filter fluorescence (upper panel) and ph1 (lower panel), scale bar = 100 μm.

**Figure 2 cells-11-02799-f002:**
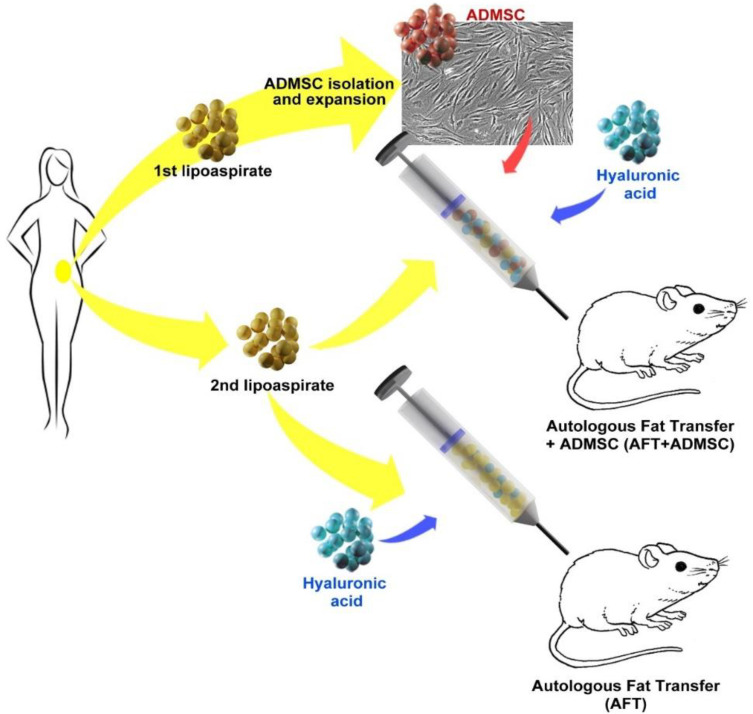
Transplantation approach: Engraftment in vivo was performed according to the autologous fat transfer (AFT) method with/without cells. The AD-MSCs were isolated and expanded from a first lipoaspirate and combined with autologous fat (second lipoaspirate) obtained from the same patient; for the in vivo engraft for cell tracking studies, AD-MSCs were genetically modified by GFP viral vector (AD-MSC-GFP); AFT alone was performed as a control by implantation of only lipoaspirate derived from the same donor, and hyaluronic acid was added as a carrier.

**Figure 3 cells-11-02799-f003:**
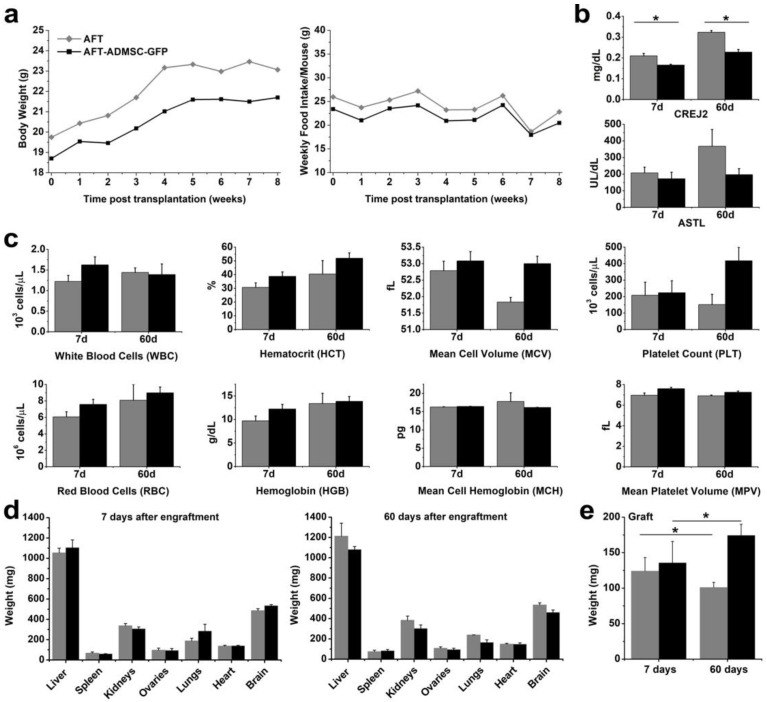
Xenotransplantation did not affect the NOD SCID mice’s health: (**a**) the average weight of the mice (left graph) and food intake (right graph) were similar between the two groups; (**b**) serological CREJ2 increased in the AFT group compared to the AFT-ADMSC-GFP group, while the ASTL levels were similar; (**c**) the hematological parameter results were similar for both groups; (**d**) the organ weight results were comparable between the AFT and AFT-ADMSC-GFP groups at 7 days (graph left) and at 60 days (right graph); (**e**) the grafts had similar weights at 7 days post-transplantation, while at 60 days, AFT-ADMSC-GFP had a statistically significant increase compared to AFT. * *p* < 0.05, *n* = 7; grey = AFT group; black = AFT-ADMSC-GFP group.

**Figure 4 cells-11-02799-f004:**
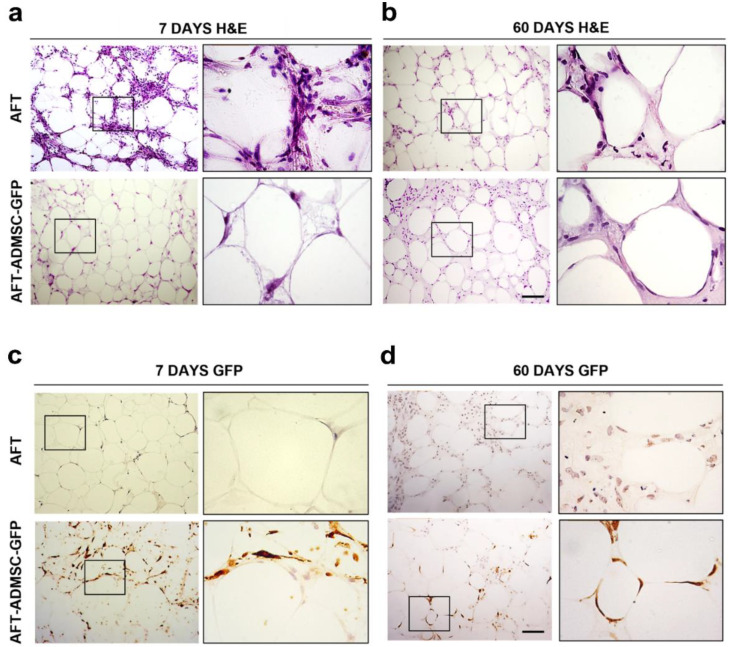
AD-MSC-GFP were tracked in the graft at 7 and 60 days after transplantation. (**a**,**b**) H&E-stained representative photomicrographs of AFT (upper panel and inset) and an AFT-ADMSC–GFP specimens (bottom panel and inset) at 7 days post-engraftment and 60 days post-engraftment, respectively. (**c**) Immunohistochemistry of GFP staining of specimen (upper panel and inset) and AFT-ADMSC-GFP specimens (bottom panel and inset) at 7 days post-engraftment; the AFT samples were negative for GFP staining. On the contrary, the AFT-ADMSC-GFP positive cells were tracked in the thin layer of connective tissue among the adipocytes cluster. (**d**) Immunohistochemistry GFP staining of AFT (upper panel and inset) and AFT-ADMSC-GFP specimens (bottom panel and inset) at 60 days post-engraftment. The AFT samples were negative for GFP staining. On the contrary, the AFT-ADMSC-GFP specimens were positive. Scale bar = 100 µm.

**Figure 5 cells-11-02799-f005:**
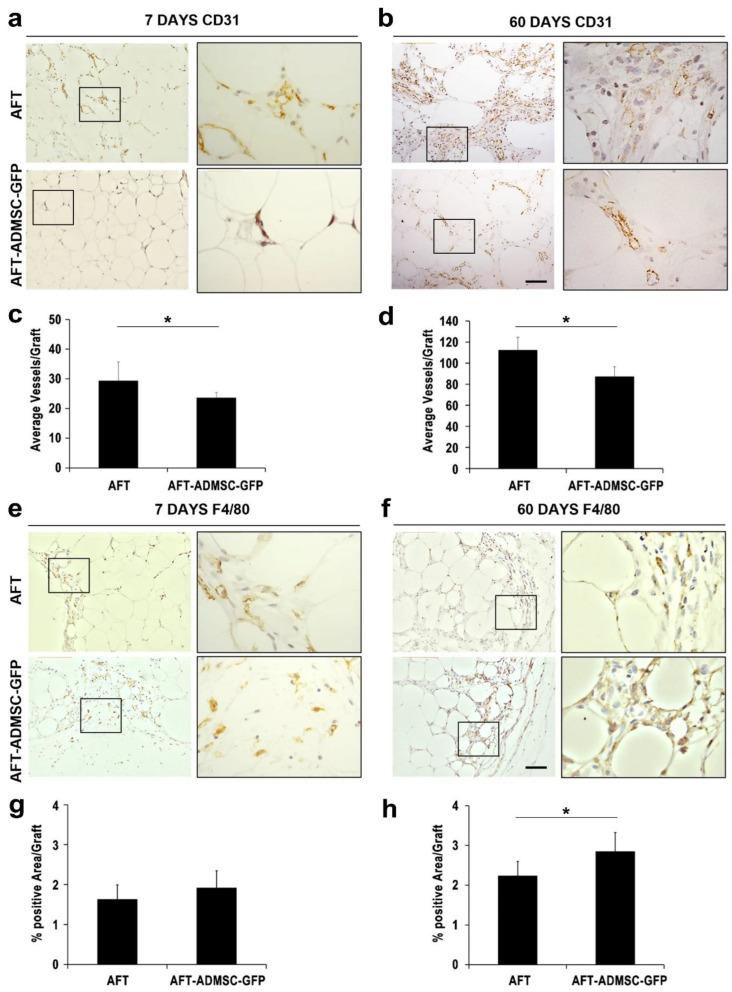
Proangiogenic and pro-inflammatory characterization of AFT-ADMSC-GFP: (**a**,**b**) immunohistochemistry for CD31 in the AFT samples (upper panel and inset) and in the AFT-ADMSC-GFP grafts (bottom panel and inset) at 7 days and 60 days after transplantation; (**c**,**d**) quantitative measurement of CD31-positive vessels for the AFT and AFT-ADMSC-GFP specimens, * *p* < 0.05; (**e**,**f**) immunohistochemistry for F4/80 in the AFT samples (upper panel and inset) and in the AFT-ADMSC-GFP grafts (bottom panel and inset) at 7 days and 60 days after transplantation; (**g**,**h**) quantification of the F4/80-positive area/graft, respectively, at 7 days and 60 days post-engraftment performed using Image J software (NIH), * *p* < 0.05; *n* = 7. Scale bar = 100 µm.

**Figure 6 cells-11-02799-f006:**
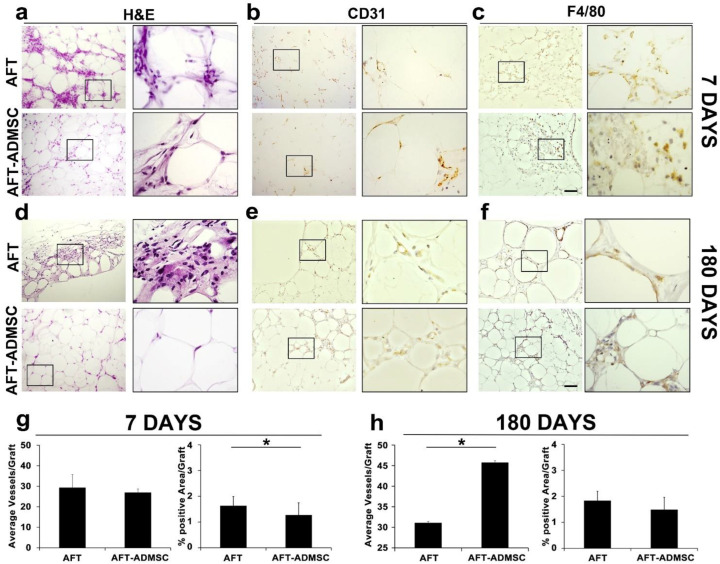
Wild-type ADMSCs exerted proangiogenic and anti-inflammatory effects on the grafts: (**a**–**c**) histology analysis of the graft at 7 days after transplantation of AFT (upper panel and inset) and AFT-ADMSC (bottom panel and inset) for (**a**) H&E, (**b**) immunohistochemistry for CD31, and (**c**) immunohistochemistry for F4/80; (**d**–**f**) histology analysis of the graft at 180 days after transplantation of AFT (upper panel and inset) and AFT-ADMSC (bottom panel and inset) for (**d**) H&E, (**e**) immunohistochemistry for CD31, and (**f**) immunohistochemistry for F4/80, scale bar = 100 µm; (**g**) quantitative measurement of CD31-positive vessels for AFT and AFT-ADMSC (left graph) and F4/80-positive area/graft (right graph) at 7 days post-transplantation; (**h**) quantitative measurement of CD31-positive vessels for AFT and AFT-ADMSC (left graph) and F4/80-positive area/graft (right graph) at 180 days post-transplantation. Quantification was performed using Image J software; * *p* < 0.05, *n* = 7.

## Data Availability

Not applicable. The dataset generated and/or analyzed during the current study are available from the corresponding author on reasonable request.

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
