# Peer review of "Human Adipose Mesenchymal Stromal/Stem Cells Improve Fat Transplantation Performance"

_cells, 2022, doi:10.3390/cells11182799_

Round 1

Reviewer 1 Report

I found very promising the possibility to enrich the engrafting AT with isolated SVF in order to increase the engraftment rate.  I suggest to investigate procedure that allows to enrich the engrafting AT with SVF for clinical practice .

Author Response

I found very promising the possibility to enrich the engrafting AT with isolated SVF in order to increase the engraftment rate.  I suggest to investigate procedure that allows to enrich the engrafting AT with SVF for clinical practice.

R: We thank very much the Referee for the very positive feedback, we also appreciate the suggestion. We have therefore improved the introduction section (lines 57-66) describing the use of the stromal vascular fraction as additive for adipose tissue-derived mesenchymal stem cells in clinical practice.

Reviewer 2 Report

The authors reported a xenograft model of adipose tissue engineering in which NOD/SCID mice underwent autologous fat transfer (AFT) with/without human adipose derived mesenchymal stromal stem cells (AD-MSCs) and were monitored for 7, 60, and 180 days. The result indicated the protective role of adipose progenitors in autologous AFT tissue resorption.

 The work contributes to the understanding of the role of AD-MSCs involved in AFT mentioned above. However, more data and details must be included, in order to really provide robust data that may support the authors' claims.

 1.    Animal grouping. “The animals were divided into 3 groups: 2 groups (AFT-AD-MSC and AFT-AD-MSC-GFP) received an autologous fat graft supplemented by AD-MSC naïve or AD-MSC-GFP, and hyaluronic acid (HA), respectively, while a third group received only an autologous fat graft (AFT) as a control.” In the AFT-AD-MSC group and AFT-AD-MSC-GFP group, the AD-MSC were suspended in 40μL of carrier solution composed of 25% human serum AB (hsAB), 25% HA hydrogel, 50% saline solution, and then 200 μL of human autologous lipoaspirate were added. However, in the 3rd group, only the human autologous lipoaspirate were used. The carrier solution was not used in the third group, it may affect the experiment result.  Authors need to add one more group: AFT+carrier solution.

“AFT-AD-MSC” sometimes was described as “AD-MSC-AFT” , please revise.  For example, line 155, line 182.

2.    The animals sacrifice time points are need to consider. 7 and 60 days post-transplant for the AFT-AD-MSC-GFP group, 7- and 180-days post-transplant for the AFT- AD-MSC group; and 7, 60, and 180 days post-transplant for the AFT control group. Why there is no time point of 180 days in the AFT-AD-MSC-GFP+ group, and 60 days after transplantation for the AD-MSC-AFT group? To compare the difference between AFT-AD-MSC-GFP group and AFT- AD-MSC group, they should have the same time points for collecting samples. Otherwise, its difficult to interpret the inflammation and angiogenesis.

3.    Line 304, The authors describe that “both AFT and AFT-AD-MSC- show a thin layer of connective tissue in the external area and a central core predominantly composed of adipocyte clusters immersed in extracellular structure”. Please use arrows to show concreted structure, such as, the “thin layer of connective tissue” in the external area, and “a central core”. If the authors have figures of the collected samples, which will more helpful.

4.    Many citing reference were lost in the manuscript, please add the corresponding reference. There are also lots of mistakes in the manuscript. For example, line 104, “H2O should be H2O”, line 120, CO2, line 129, (1 × 106 transducing units/mL), etc. Please review the manuscript carefully and revise them.

5.    Please add the “n” value in the figure legend.

6.    In Figure 3e, Figure 5c, d, h and Figure 6 g, h, there are only two groups in each figure, however, each column has one *, I don’t understand. Please explain the reason and revise them.

Author Response

The authors reported a xenograft model of adipose tissue engineering in which NOD/SCID mice underwent autologous fat transfer (AFT) with/without human adipose derived mesenchymal stromal stem cells (AD-MSCs) and were monitored for 7, 60, and 180 days. The result indicated the protective role of adipose progenitors in autologous AFT tissue resorption.

The work contributes to the understanding of the role of AD-MSCs involved in AFT mentioned above. However, more data and details must be included, in order to really provide robust data that may support the authors' claims.

Animal grouping. “The animals were divided into 3 groups: 2 groups (AFT-AD-MSC and AFT-AD-MSC-GFP) received an autologous fat graft supplemented by AD-MSC naïve or AD-MSC-GFP, and hyaluronic acid (HA), respectively, while a third group received only an autologous fat graft (AFT) as a control.” In the AFT-AD-MSC group and AFT-AD-MSC-GFP group, the AD-MSC were suspended in 40μL of carrier solution composed of 25% human serum AB (hsAB), 25% HA hydrogel, 50% saline solution, and then 200 μL of human autologous lipoaspirate were added. However, in the 3rd group, only the human autologous lipoaspirate were used. The carrier solution was not used in the third group, it may affect the experiment result.  Authors need to add one more group: AFT+carrier solution.

R: We appreciated referee comment and agree with the need of clarity on the carrier. This was added in all the groups, we apologize if this was not clear. Details have been added in line 177 within the material and methods section. Moreover, figure 2 has been updated accordingly.

“AFT-AD-MSC” sometimes was described as “AD-MSC-AFT”, please revise.  For example, line 155, line 182.

R: Agreed. The text has been revised accordingly.

  1. The animals sacrifice time points are need to consider. 7- and 60-days post-transplant for the AFT-AD-MSC-GFP group, 7- and 180-days post-transplant for the AFT- AD-MSC group; and 7-, 60-, and 180-days post-transplant for the AFT control group. Why there is no time point of 180 days in the AFT-AD-MSC-GFP+ group, and 60 days after transplantation for the AD-MSC-AFT group? To compare the difference between AFT-AD-MSC-GFP group and AFT- AD-MSC group, they should have the same time points for collecting samples. Otherwise, its difficult to interpret the inflammation and angiogenesis.

R; Please allow us to explain better:

  1. The 7 and 60-days post-transplant for the AFT-AD-MSC-GFP group was intended to track implanted cells considering that several in vivo studies demonstrated that GFP MSC modified persist up to 8 weeks (Guo Y et al 2012, Wei N et al 2021, Duan X et al 2017).
  2. The 7- and 180-days post-transplant for the AFT- AD-MSC group was intended to observe early and very late impact of AD-MSC in AFT performance by immunohistochemical studies.
  3. The 7-, 60-, and 180-days post-transplant for the AFT control group was intended as a reference group to observe early and very late impact of AFT alone performance (for group 2), introducing an additional time point (60 days) to have an intermediate time point to be compared with group 1.

While we somehow agree with Referee comment, we have been also limited in the experimental design by local ethical committee and Italian ministry of the health for the introduction of further animal groups (AD-MSC GFP at 180 days, AD-MSC at 60 days). This was planned also considering the evidence that GFP labelling does not modify MSC biological activities (Guo et al 2012, Wei N et al 2021), indicating that GFP could be used both to track MSC distribution and monitor their impact in AFT compared to AFT alone (60 days).  

  1. Line 304, The authors describe that “both AFT and AFT-AD-MSC- show a thin layer of connective tissue in the external area and a central core predominantly composed of adipocyte clusters immersed in extracellular structure”. Please use arrows to show concreted structure, such as, the “thin layer of connective tissue” in the external area, and “a central core”. If the authors have figures of the collected samples, which will more helpful. 

R: To encounter referee suggestion, we improved the manuscript by adding supplementary figure 1A-D  of the AFT and AFT-AD-MSC-GFP samples at 7 days and 60 days after transplantation. AxioZoomV16 was used to acquire lower magnification images (objective 1.5X with digital magnification 270X) where the thin layer of connective capsula and central core are better represented

  1. Many citing reference were lost in the manuscript, please add the corresponding reference. There are also lots of mistakes in the manuscript. For example, line 104, “H2O should be H2O”, line 120, CO2, line 129, (∼1 × 106 transducing units/mL), etc. Please review the manuscript carefully and revise them.

R: We apologize for this. A revised text with adjourned references is now provided.

  1. Please add the “n” value in the figure legend. 

R: The n value (n=7) was added in the legend of the figures 3,5, 7

  1. In Figure 3e, Figure 5c, d, h and Figure 6 g, h, there are only two groups in each figure, however, each column has one *, I don’t understand. Please explain the reason and revise them.

R: Thank you for pointing this out. We agree, the asterisk shall be present only on one bar and not on two. Figures have now been corrected.

Reviewer 3 Report

Resorption of fat graft is a major issue in autologous fat transfer. In this manuscript, the authors investigated the resorption of human fat graft enriched or not with adipose stromal cells (ADSCs), after transplantation in an immune-deficient mouse model. They conclude that ADSCs protect against resorption thanks to their anti-inflammatory and proangiogenic activities.  

 My major concerns are the following:

h- The  lack of originality:  Several papers have previously demonstrated in experimental and clinical research the efficacy of fat grafts enriched with adipose stromal cells. Authors did not enough mention these papers. As an example, see the review: “Cell-Assisted Lipotransfer: A Systematic Review of Its Efficacy” by Navid Mohamadpour Toyserkani et al. 2016, or “Enrichment of autologous fat grafts with ex-vivo expanded adipose tissue-derived stem cells for graft survival: a randomised placebo-controlled trial” by Kolle et al. The lancet 2013). Still some issues have to be addressed to improve this approach, but they are not investigated in the submitted manuscript.

-     - Authors claim that “these findings confirm that it is possible to isolate ADSCs from a small (600 mg) amount of fat" (lines 208-239). I think these data are not original and should be indicated in the Methods part or in supplemental.

-         - Data showing that ADSCs differentiate into adipocytes within 60 days after transplantation are difficult to evaluate using immunohistochemistry. Another approach, more quantitative should be performed.

-          - The proangiogenic and anti-inflammatory activities of ADSCs are evaluated by mmunohistochemistry and analysed by Image J software. A complementary quantitative method should be used

-        -  It is not clear if the hyaluronic acid was added in the two groups of mice?

 Minor remarks: A careful editing should be done, as the number of the reference is missing in several brackets, and some text of the sample remain (line 406).  

T

Author Response

Resorption of fat graft is a major issue in autologous fat transfer. In this manuscript, the authors investigated the resorption of human fat graft enriched or not with adipose stromal cells (ADSCs), after transplantation in an immune-deficient mouse model. They conclude that ADSCs protect against resorption thanks to their anti-inflammatory and proangiogenic activities.  

 My major concerns are the following:

h- The lack of originality:  Several papers have previously demonstrated in experimental and clinical research the efficacy of fat grafts enriched with adipose stromal cells. Authors did not enough mention these papers. As an example, see the review: “Cell-Assisted Lipotransfer: A Systematic Review of Its Efficacy” by Navid Mohamadpour Toyserkani et al. 2016, or “Enrichment of autologous fat grafts with ex-vivo expanded adipose tissue-derived stem cells for graft survival: a randomised placebo-controlled trial” by Kolle et al. The lancet 2013). Still some issues have to be addressed to improve this approach, but they are not investigated in the submitted manuscript.

R: This Referee comment allows us to give better value to our data compared to the mentioned manuscripts

  1. Cell assisted lipotransfer (CAL) represents a strategy of fat grafting enhancement where ex vivo expanded AD-MSC or SVF are added to autologous fat. Several clinical trials have been performed by enriching AFT with SVF to treat lipodystrophic diseases such as hemifacial atrophy, craniofacial microsomia and for breast reconstruction (Coleman 2006). However, as reported by Toyserkany et al., CAL with SVF has limitation concerning the dose/volume of cells transplanted, possibly limiting the reproducibility of the procedure. On contrary, a CAL with a defined number of AD-MSC, as we are here proposing, addresses this weakness allowing the establishment of more defined ration between AD-MSC doses, fat volume and route of administration, possibly impacting on resorption and volume loss of the graft.
  2. Compared with the milestone paper of Kolle et al, we give evidence of cell tracking study (not feasible in humans) where implanted cells can be followed 60 days form implant, additionally prolonging observation time of the graft up to 180 days for histological analyses (it was limited to 120 days in the clnical trial).
  3. More, Kolle et al have conducted a crucial randomized clinical trial comparing the survival of CAL enriched with autologous AD-MSC versus regular AFT. Ten patients were enrolled and injected in the upper arm with a total volume of 34 ml of AFT including 4 ml of solution with 6.5x108 AD-MSC or with equal volume of placebo/mock solution as control. Patients were monitored for 120 days and the undergone to MRI and graft collection by surgical procedure for histological analysis (necrosis, connective tissue, angiogenesis). Although the described procedure of AD-MSC enriched fat grafting showed a good feasibility and safety profile, the study may be suitable only for major surgery where large amount of tissues are required. Therefore, that approach may be limited in microsurgery (i.e facial defect) requiring a very low amount of fat grafting, as we have been implementing using 0.250 ml of fat plus AD-MSC.
  4. Kolle et al administrated a very high dose of AD-MSC: 20x106 of AD-MSC for each ml of fat. On contrary, we select a dose of 4 times lower (5x106 AD-MSC/ml of fat), highlighting a more favourable cell performance within the graft. This has also implication in cell-based therapeutics perspective that shall require less time and lower costs for GMP manufacturing. Thus, our data indicate that a lower AD-MSC dose can exert a therapeutic benefit in a humanized animal model.
  5. In addition, we demonstrate that a carrier solution with hyaluronic acid used to embed cells and fat can be used in graft engineering, as previously demonstrated in rabbits by our group (Piccinno et al. 2013).

Collectively, while we acknowledge that amount of previously reported data, we presume that our findings have sufficient levels of originality to further support the implementation of more standardized procedures in adipose tissue regeneration. These considerations have included in manuscript Discussion (lines 522-526).

-     - Authors claim that “these findings confirm that it is possible to isolate ADSCs from a small (600 mg) amount of fat" (lines 208-239). I think these data are not original and should be indicated in the Methods part or in supplemental

R: To encounter referee suggestion, we moved the details about the starting amount of fat from the Results to Materials and Methods section (please see line 86).        

-         - Data showing that ADSCs differentiate into adipocytes within 60 days after transplantation are difficult to evaluate using immunohistochemistry. Another approach, more quantitative should be performed.

R: We agree with the reviewer and, to encounter this need, we assessed differentiation by a quantitative approach using advanced microscopy. Axiozoom V16 (Zeiss) equipped with ZenPro software (Zeiss) was used to acquire pictures at low magnification (1.5X with digital magnification of 270X). Multiple (n=5) images were captured from AFT groups and AFT-MSC-GFP at 60 days after transplantation. The number of adipocytes was scored, resulting an average value (± SD) of 28.4 (±7.6) and 45.6 (±6.73) in AFT and AFT-AD-MSC-GFP respectively. The addition of AD-MSC-GFP increase the adipocytes number in a statistically significant manner (p-value < 0.01) with a fold of increase equal to 1.6.

In addition, the area corresponding to adipocytes was automatically quantified by ZenPro Software resulting in an average area (± SD) of 9280.66 ±1957.6 µm2 and 13465.71±1398.53 µm2 in AFT and AFT-AD-MSC-GFP, respectively. AFT-AD-MSC-GFP groups have a statistically significant increase of adipose tissue area (p-value<0.01). These results further support the evidence that AD-MSC treated mice retain a higher differentiation potential in comparison AFT alone.  

These data are now added in Material and Method section (lines 229-233), in result section (lines 361-371) and in the supplementary figure 1E-F.

-          - The proangiogenic and anti-inflammatory activities of ADSCs are evaluated by immunohistochemistry and analysed by Image J software. A complementary quantitative method should be used

To follow the referee suggestion, the anti-inflammatory and proangiogenic properties of AD-MSC were assessed accounting that angiogenesis and inflammation are co-dependent processes. Some forms of inflammation, especially chronic inflammation, can stimulate vessel growth, contributing to a tissue’s altered inflammatory response (Walsh Da et al, 2001). PGE2 is the major prostaglandin generated by COX and has been implicated in the anti- inflammatory response though M2 macrophages differentiation promoting a resolution phase, increasing phagocytic activity and decreasing inflammation (Lu D et al 2021). PGE2 is involved also in angiogenesis regulation (Nemeth K et al 2009; Choi H et al 2011; Finetti F et al, 2007). Zhang et al, 2011 investigate by an in vitro and in vivo studies that PGE2 acted directly on endothelial cells binding 4 cognate receptors named EP1, EP2, EP3, and EP4 that belong to the G protein–coupled receptor superfamily, promoting assembly of new blood vessels through selective activation of PKACγ signal pathway (Zhang et al 2011). Thus, we focused PGE2, as main player of inflammation/angiogenesis. These data are now described in the Materials and Methods (lines 221-223) / Results sections and reported in Supplementary Figure 2.

First, we in vitro stimulated AD-MSC by LPS (8ug/ml) for 24 hours as described (Kurte M et al 2020; Lin T et al 2017). The levels of secreted PGE2 were measured and analyzed in comparison with unstimulated cells. After 24 hours of LPS administration, MSC secreted significantly higher levels of PGE2 (13.062 ± 4.45 pg/ml pg/ml if compared with unstimulated cells (8.023 ± 1.18 pg/ml; p-value <0.01 in Supplementary Figure 2A).

Moreover, we took advantage of referee suggestion and we investigated PGE2 levels in stored mouse sera to  ultimately quantified their levels in AFT-AD-MSC group in comparison with AFT only treated mice. Frozen stocks of animal sera were defrosted and analyzed. Very interestingly, we found that the circulating levels of PGE2 detected in AFT and in AFT-AD-MSC treated mice were respectively 27.95±11.35 pg/ml and 54.78 ±7.76 pg/ml suggesting a role of PGE2 in AD-MSC anti-inflammatory/pro-angiogenic effect (p-value<0.001; Supplementary Figure 2B).

PGE2 related experiments and data were added in the manuscript text: lines 139-245 and 221-223 material and method section; lines 471-481 results section and lines 569-579 in the discussion section.

-      It is not clear if the hyaluronic acid was added in the two groups of mice?

      R: We apologize if this has not been clear. HA was used in all groups. Material and Methods and Figure 2 have been updated to clarify it.

 Minor remarks: A careful editing should be done, as the number of the reference is missing in several brackets, and some text of the sample remain (line 406).  

R: We are sorry for this. Text has been revised and references added in the empty brackets.  

Round 2

Reviewer 2 Report

The authors addressed all the questions. The manuscript can be accepted.

Reviewer 3 Report

The authors fully addressed all my concerns, and greatly improved  the quality of the manuscript.